**Cite this article:** Kennedy GM. 2020
The unexpected narrowness of eccentric debris
rings: a sign of eccentricity during the
protoplanetary disc phase. *R. Soc. Open Sci.* **7**:
200063.

astrophysics/extrasolar planets

circumstellar matter, debris discs,
protoplanetary discs, planet–disc interaction

**Author for correspondence:**
Grant M. Kennedy
e-mail: g.kennedy@warwick.ac.uk

# The unexpected narrowness of eccentric debris rings: a sign of eccentricity during the protoplanetary disc phase

## Grant M. Kennedy[1,2]

[1]Department of Physics, and [2]Centre for Exoplanets and Habitability, University of Warwick, Gibbet Hill Road, Coventry CV4 7AL, UK

GMK, 0000-0001-6831-7547

This paper shows that the eccentric debris rings seen around the stars Fomalhaut and HD 202628 are narrower than expected in the standard eccentric planet perturbation scenario (sometimes referred to as 'pericentre glow'). The standard scenario posits an initially circular and narrow belt of planetesimals at semi-major axis $a$, whose eccentricity is increased to $e_f$ after the gas disc has dispersed by secular perturbations from an eccentric planet, resulting in a belt of width $2ae_f$. In a minor modification of this scenario, narrower belts can arise if the planetesimals are initially eccentric, which could result from earlier planet perturbations during the gas-rich protoplanetary disc phase. However, a primordial eccentricity could alternatively be caused by instabilities that increase the disc eccentricity, without the need for any planets. Whether these scenarios produce detectable eccentric rings within protoplanetary discs is unclear, but they nevertheless predict that narrow eccentric planetesimal rings should exist before the gas in protoplanetary discs is dispersed. PDS 70 is noted as a system hosting an asymmetric protoplanetary disc that may be a progenitor of eccentric debris ring systems.

## 1. Introduction

Debris discs are the circumstellar discs that are seen to orbit main-sequence stars, including our Sun. The dust that is observed derives from a parent population of planetesimals; around other stars this replenishment is inferred because the lifetime of the dust in the presence of radiation and stellar wind forces is typically shorter than the stellar age [1], while in the Solar system the connection is more direct because both the dust and parent bodies are detectable [2,3].

In the Solar system, the structure of the asteroid and Edgeworth–Kuiper belts is shaped by planets. Indeed, these small-body populations have arguably contributed far more per unit mass to our understanding of Solar system history than the planets themselves. Perhaps the most famous example is the inference of Neptune's outward migration from Pluto's capture into the 2 : 3 mean-motion resonance [4].

One key challenge for debris disc science is the successful application of similar concepts to other stars. While a future aspiration is to unravel the histories of other planetary systems [5], one past and present goal is to correctly infer the presence of as-yet undetectable planets via their gravitational influence [6,7]. Because these perturbations tend to act on timescales that are longer than dust lifetimes, the general expectation is that the planetary influence is imprinted on the parent planetesimal population and inherited by the collisional fragments that are observed.

Connecting disc structures, of which myriad are seen, to planets, has proven hard, primarily because detecting the putative planets is hard. Indeed, it is normally impossible to rule out the proposed planets, partly because they commonly lie somewhere along a locus in mass–semi-major axis parameter space rather than in specific locations, but primarily because their masses can be far too small for detection. The single successful example of prediction and subsequent detection is for the edge-on disc $\beta$ Pictoris, whose warp was attributed to an inclined planet [8] that was subsequently discovered by direct imaging [9,10].

While $\beta$ Pic b was predicted based on the long-term 'secular' perturbations from a misaligned planet, warps in discs are normally hard to detect because most systems do not have the optimal edge-on geometry. In-plane perturbations that result in azimuthally dependent structures are more generally detectable, typically manifesting as a significant eccentricity, which may be imaged directly or inferred from a brightness asymmetry at lower spatial resolution. The first detection of an eccentric debris disc was for HR 4796 (which was in fact also the second debris disc to be imaged [11,12], the first being $\beta$ Pic), where a brightness asymmetry (the so-called 'pericentre glow') from low-resolution mid-infrared imaging was attributed to an unseen planet [6,13].

Despite a high sensitivity to asymmetry (due to the exponential dependence of flux density on dust temperature), mid-IR pericentre glow is rarely the method by which asymmetric discs are identified, simply because few discs are bright enough to be imaged near 10 µm (both in absolute terms, and relative to the host star). Instead, scattered light images have proven far more successful and yielded a veritable zoo of structures [14–17]. While many of these images suggest the influence of unseen planets (more systems than might be surmised based on a simple expectation of eccentric rings, [7]), a major limitation is that these images trace small dust. This dust is subject to strong radiation and stellar wind forces, and possibly gas drag which, in addition to opening the possibility of entirely different sculpting scenarios [18,19], makes connecting the observed structure to the orbits of the underlying parent planetesimals difficult. Ideally, inferences of unseen planets would be made at longer wavelengths, where the typical grain sizes are large enough to be immune to non-gravitational perturbations, and the observed structure more reasonably assumed to be representative of the planetesimal orbits.

Until recently, millimetre-wave observation of debris discs was limited to photometry and marginally resolved imaging [20,21]. However, the unprecedented sensitivity and resolution of the Atacama Large Millimetre/Submillimetre Array (ALMA) means that well-resolved debris disc images are now of sufficient quality to be confronted with dynamical and collisional models. Indeed, after considering the motivation in more detail (§2) this paper shows that ALMA images of the debris discs in the Fomalhaut and HD 202628 systems have a narrower radial width than is expected based on the secular perturbation scenario originally devised to explain pericentre glow for HR 4796 (§3). Possible origins, including the possibility that these eccentric discs are actually a relic from the protoplanetary disc phase, are discussed in §4, and concluding remarks made in §5.

## 2. Motivation and expectations

The basic idea that underpins this work is that the width of any debris ring provides information on the eccentricities of the objects that are observed. An axisymmetric debris ring's width could be entirely explained by orbits of a single semi-major axis $a$ and non-zero eccentricity, as long as the pericentre angles are uniformly distributed in azimuth. The eccentricities could, however, be lower, because some (or all) of the ring width could also arise from a range of semi-major axes. Thus, in most cases one expects to be able to derive an upper limit on the eccentricities. Essentially the same argument applies to eccentric debris rings, but the constraint is a bit more complex because the pericentre angles must have a preferred direction to break the symmetry. To understand this constraint and why it is

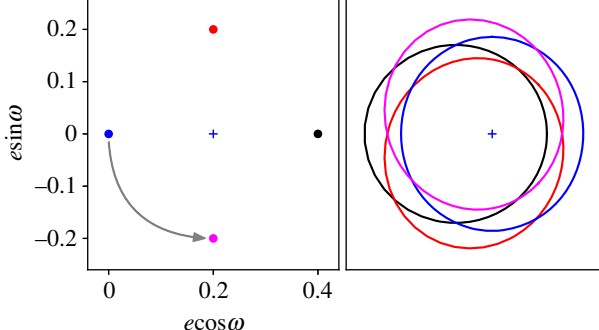

**Figure 1.** Illustration of how secular perturbations produce an eccentric debris ring, with $e\omega$-space in the left panel, and particle orbits viewed from above in the right panel. Particles that initially have zero eccentricity precess anticlockwise around the forced eccentricity (here $e_f = 0.2$, $\omega = 0$ is marked by the blue + symbol). A precessing particle starting at the blue dot (i.e. with $e_p = 0.2$) would pass through the magenta dot, then the black dot, then red, then blue, etc. The orbits for these four points in the precession cycle are shown in the right panel, where the star is marked by the blue + symbol. In particular, note that the black orbit is eccentric while the blue one is circular. The sum of many such orbits distributed in a circle around $e_f$ produces an eccentric ring (see upper right panel in figure 2).

useful, this section briefly describes the secular perturbation mechanism that is supposed to be the origin of eccentric debris rings (see [6] for a detailed explanation), and how particle eccentricities and pericentres can be related to observable disc structure. These ideas form the basis for the model used below.

Secular (long-term) perturbations from an eccentric planet cause disc particles' pericentre angles to precess, during which their eccentricities vary in a systematic way. As applied to debris discs, this scenario assumes that it is the large and long-lived planetesimals that are perturbed onto eccentric orbits, and that smaller bodies inherit these eccentric orbits when they are created. This orbital evolution is best visualized in eccentricity–pericentre phase space (here called $e\omega$-space, but $hk$-space is also used), where the distance from the origin is the eccentricity, and the anti-clockwise angle from the positive $x$-axis the pericentre direction relative to some reference direction (normally the ascending node). The planet imparts a 'forced' eccentricity $e_f$ on test particles, whose magnitude is approximately the planet/particle semi-major axis ratio (assuming an interior planet) multiplied by the planet eccentricity, and whose pericentre direction is the same as the planet's. A test particle with the forced eccentricity stays on this orbit, but particles elsewhere in $e\omega$-space move continuously in an anticlockwise circle around the forced eccentricity (i.e. their eccentricity changes as they precess). The precession rate depends on the strength of the planet's perturbation, via the planet mass and planet/planetesimal semi-major axis ratio. An illustrative example for a particle with zero initial eccentricity is given in figure 1, which shows the particle and the corresponding orbits at four points in the precession cycle. The difference between a particle's actual eccentricity and the forced eccentricity is termed the 'free' or 'proper' eccentricity $e_p$, which is also the radius of the circle drawn out by the particle. The motion in $e\omega$-space is such that the particle's orbit is most eccentric only when the pericentre angle is near zero (i.e. near the black dot); this behaviour is the key to how secular perturbations produce an eccentric ring.

Thus, given an initial population of particles at some common location in $e\omega$-space, but with a finite range of semi-major axes, the long-term evolution due to secular perturbations from an eccentric planet is that these particles are spread around a circle of radius $e_p$ that encloses the forced eccentricity. The time taken for particles to spread out depends on their differential precession rates, and thus the range of semi-major axes. The case of a Gaussian distribution of near-zero initial eccentricities with dispersion[1] $\sigma_{e,p} = 0.01$ is shown in the upper right panel of figure 2, with the initial eccentricities as orange dots, and the final eccentricities after many precession cycles as blue dots (the semi-major axes are distributed as a narrow Gaussian with dispersion $\sigma_a/a = 0.01$). In this case, $e_p \approx e_f$, the full range of eccentricities is $\approx 2e_f$, and the resulting debris ring has a uniform width of $\approx 2ae_f$ (looking back at figure 1 may help make these results clear). Alternatively, if the initial eccentricities were very close to

---

[1]In this work, the word 'dispersion' is used to describe the width of a Gaussian distribution, while 'range' refers to the full range of values covered by some parameter. So in the upper right panel of figure 2, the initial eccentricity dispersion is small, but the range of final eccentricities is large. Below, both Gaussian and top-hat semi-major axis distributions are used, in which case 'range' is used to describe the width of either distribution.

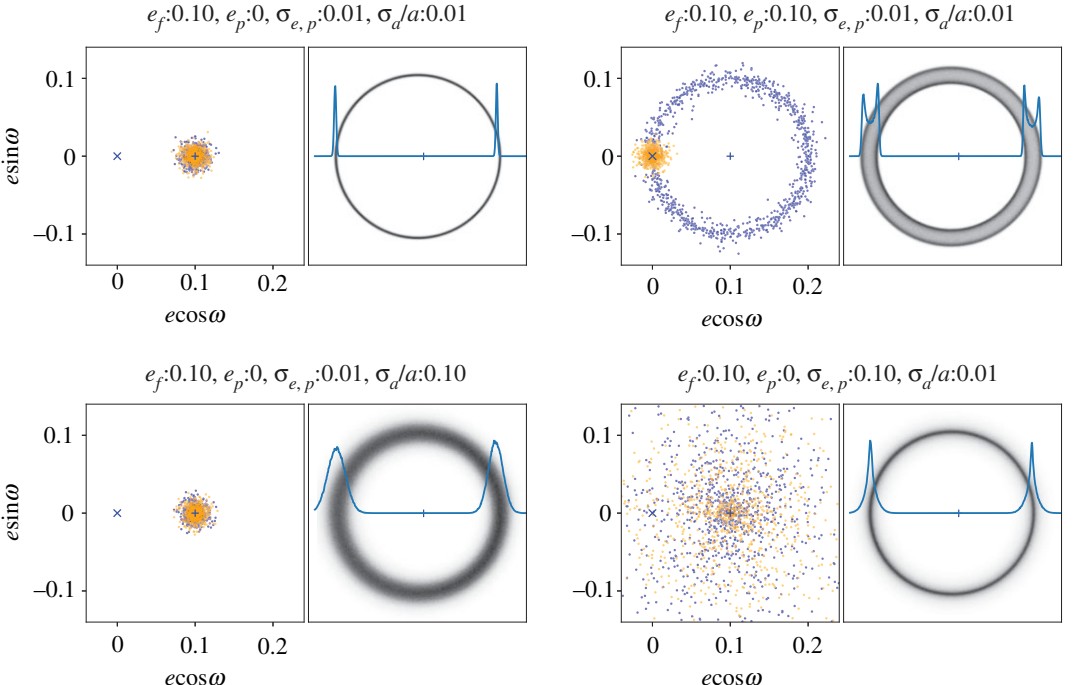

**Figure 2.** Single-planet secular perturbations with different initial eccentricities. In each panel, the left plot shows the $e\omega$-space (initial, orange; final, blue), and the right image shows the resulting eccentric disc image, with the star marked by the blue + (assuming face-on geometry, the stated range of semi-major axes, and including a $1/\sqrt{r}$ weighting to account for the radially varying surface brightness at long wavelengths). The line in the right image shows the radial profile along the pericentre direction. The forced eccentricity is the same in all panels. The top left panel shows a narrow ring, while the other three show how the disc width can arise from small initial eccentricities (top right), a large initial eccentricity dispersion about the forced eccentricity (bottom right), or a range of semi-major axes (bottom left). Only a range of semi-major axes yields an azimuthally varying ring width.

$e_f$, as shown in the upper left panel of figure 2, the final eccentricities are also near $e_f$, and the resulting debris ring appears very narrow (i.e. approximately follows a single eccentric orbit). Naively, one expects that the initial planetesimal eccentricities are small, the result of damping during the gas-rich protoplanetary disc phase, so the former of these two cases seems more physically plausible [22]. That is, with this model one expects eccentric debris rings to have widths of $\approx 2ae_f$, and narrower widths would imply non-zero initial eccentricities that are shifted towards the forced eccentricity.

Further variations on this model are of course possible; the semi-major axis distribution can be made wide enough to dominate the belt width, in which case the width varies as a function of azimuth (e.g. with $\sigma_a/a = 0.1$ and $e_p = 0$; see the lower left panel of figure 2). The dispersion of the initial eccentricity distribution can also be varied; as $\sigma_{e,p}$ increases, the radial disc profile acquires a lower level 'halo' that is not present when the dispersion is small (compare the lower right and upper left panels in figure 2). Thus, $e_p$ changes the width, and $\sigma_{e,p}$ the radial concentration of an eccentric ring, and the semi-major axis distribution provides a further means to change the width, but in a way that is azimuthally dependent. While alternative parametrizations could also describe an eccentric disc, it could not achieve the same flexibility with fewer parameters, and the advantage here is that the model is physically connected to the initial conditions via the assumed secular perturbation scenario. Of course, whether these parameters are all needed depends on the data in hand, which is considered below.

Whether the range of semi-major axes should be small, or be distributed in a particular way is unclear. To briefly explore this aspect, figure 3 shows two examples that have the same eccentricity distributions; panel (a) uses a Gaussian distribution of semi-major axes centred at $a$ with dispersion $\sigma_a$, while (b) uses a uniform distribution between $a \pm \delta_a/2$. In terms of the resulting disc images, figure 3a is similar to the lower left panel in figure 2, with some additional width contributed by an increased range of eccentricities. Figure 3b is similar to the upper right panel in figure 2, but has two bright rings near the middle. These arise because this disc is essentially a superposition of two such rings, one at $a - \delta_a/2$ and the other at $a + \delta_a/2$. The outer of the bright rings is the outer edge of the component with $a - \delta_a/2$, and the inner ring the inner edge of the component with $a + \delta_a/2$. Perhaps

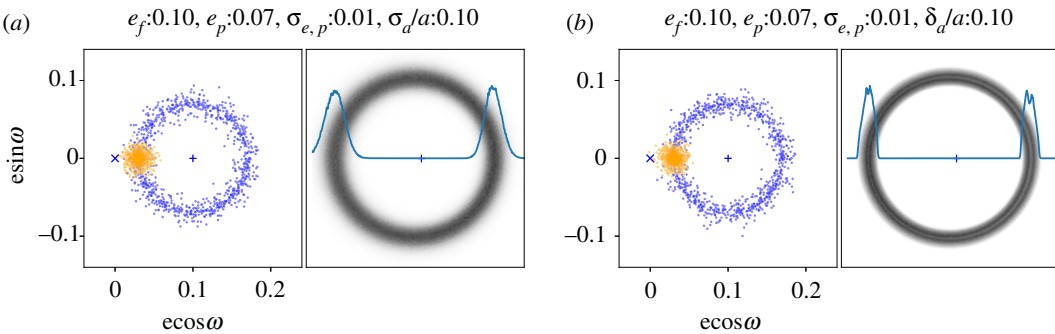

**Figure 3.** Single-planet secular perturbations with different semi-major axis distributions. Conventions are as in figure 2. The left panel is similar to the lower left panel in figure 2, but with a non-zero proper eccentricity. The right panel has the same eccentricity parameters, but the semi-major axis distribution is uniform in the range $a \pm \delta_a/2$. The double-ring structure in the right panel can be thought of as the superposition of two rings like the top right panel of figure 2 with semi-major axes $a - \delta_a/2$ and $a + \delta_a/2$. As can be seen from the radial profiles, near the peak brightness the disc is narrower at apocentre than pericentre (in contrast to all other models).

importantly, the distance between these rings is not constant with azimuth, and is actually smaller near apocentre, in contrast to all other models. Given an appropriate disc configuration, observations that moderately resolve the disc width might be able to detect or rule out such a variation. Note, however, that this example is highly idealized, with sharp edges in the semi-major axis distribution, and eccentricities that do not vary with semi-major axis.

## 2.1. Previous work

Lee & Chiang [7] developed a similar model, which was extended to incorporate radiation forces on small grains, and showed that many of the unusual scattered light morphologies observed for bright debris discs can be explained by highly eccentric parent belts ($e_f \sim 0.6$). Their work assumed a very small proper eccentricity of 0.02, thus implicitly assuming either that the parent belt was initially very eccentric, or that particles damp to the forced eccentricity before they fragment to small enough sizes that they are strongly perturbed by radiation forces. It seems unlikely that their reproduction of scattered light structures would appear as compelling with $e_f = e_p$, as the parent belts would be predicted to be much wider (the effect on smaller particles is less clear, most likely the structures would become less distinct). To take a specific example, Esposito *et al.* [23] model the HD 61005 disc and find $e_f = 0.21$ and $e_p = 0.08$, where the small proper eccentricity is required to reproduce the relatively narrow parent belt. Thus, it seems probable that if highly eccentric belts do explain the range of scattered light structures, those belts should have proper eccentricities that are smaller than is expected.

However, whether the range of observed debris disc structures can be 'unified' remains unclear, as these models predict that asymmetric structure seen in scattered light should in at least some cases be accompanied by asymmetric structure in millimetre-wave observations. Few such systems have been observed by ALMA, but HD 15115 provides a first test. The scattered light structure is highly asymmetric and can be explained with an eccentric belt with a pericentre direction in the sky plane [7], which predicts that the millimetre-wave emission should also appear asymmetric. However, MacGregor *et al.* [24] find that the structure is consistent with being symmetric, suggesting that in this case a highly eccentric parent belt is not the reason for the asymmetric scattered light structure. The potential for such stark differences shows that connecting observations with the underlying planetesimal belt structure is more easily done with thermal emission at longer wavelengths, such as probed by ALMA.

## 3. The ring widths of Fomalhaut and HD 202628

Given the motivation above, several eccentric systems are well suited to quantifying the ring width. Here Fomalhaut and HD 202628 are singled out as the best cases, because their debris rings are obviously eccentric and narrow, and because extant ALMA observations have sufficient spatial resolution to constrain the ring width to better than the expected $2ae_f$. That is, the null hypothesis is that the ring widths are consistent with that expected from secular perturbations and initially near-zero

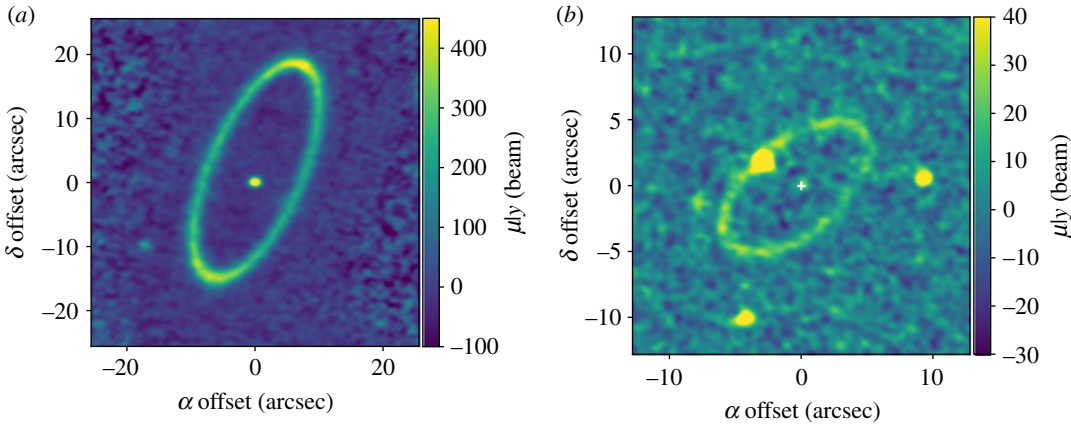

**Figure 4.** Naturally weighted clean images of the Fomalhaut (*a*) and HD 202628 (*b*) systems.

eccentricities. The assumption that the dust grains detected with ALMA share the same orbits as the planetesimals from which they are derived will be revisited below. Clean images of these discs are shown in figure 4, primarily to aid the interpretation of the residual images shown in figures 6 and 8, The Fomalhaut observations were at 1.3 mm (band 6) and have a spatial resolution of 1.6 × 1.2 arcsec resolution (12 × 9 AU at the 7.7 pc (parsec) distance of Fomalhaut). The large angular size relative to the primary beam means that seven pointings were required to cover the disc adequately. The HD 202628 observations were also at 1.3 mm and use a single pointing. The spatial resolution is 0.9 × 0.8 arcsec, which is 21 × 19 AU at the 23.8 pc distance of HD 202628. The reader is referred to the papers cited below for more observational details.

The HD 202628 ring was modelled by Faramaz *et al.* [25], and with the assumption of constant surface density (justified by a marginally radially resolved image) the width was found to be 22 AU. Given their best-fit eccentricity of 0.09 the expected width of $2ae_f = 28$ AU is larger than observed, suggesting that the proper eccentricity is lower than the forced eccentricity. The ring is at best marginally resolved radially and not detected with a high signal-to-noise ratio (s/n), but the star is detected at the expected location which significantly improves the constraints on the ring geometry. Their model was simplified in that it used an offset circular ring; here the data are modelled again using the more complex eccentric ring model. Given the data, this is not necessarily a superior approach, but has the benefit that ring parameters directly related to the secular perturbation scenario are derived explicitly and their (joint) posterior distributions quantified.

The width of the Fomalhaut ring has also been suggested to be narrower than expected by White *et al.* [26] and MacGregor *et al.* [27], with these works finding full width at half maxima (FWHM) of 13–13.5 AU. This width is to be compared to the expectation of $2ae_f = 33$ AU. However, the significance of this narrowness is not particularly clear for two reasons. Fig. 5 of MacGregor *et al.* [27] suggests that the disc has an FWHM along the major axis of approximately 5 arcsec (39 AU), but the contours in their fig. 1 show that the disc is at most only approximately 2 beams (i.e. approx. 2 arcsec) wide where it is detected (suggesting an axis labelling/scaling error). Figure 5 shows the radial profile for a naturally weighted image with a 1 arcsec wide swathe through the star and along the disc major axis. The disc widths along the disc major axis are 16 and 20 AU, with the NW ansa being narrower and accounting for the beam width of about 1.2 arcsec, these widths are consistent with a true disc FWHM of about 13 AU (i.e. much less than 33 AU).

A second issue is that the modelling in MacGregor *et al.* [27] finds a proper eccentricity (0.06 ± 0.04) that is consistent with the forced eccentricity (which was 0.12 ± 0.01). This result therefore suggests that the disc can be consistent with a width of 33 AU (4.3 arcsec) at $2\sigma$, which is clearly incompatible with the data (e.g. from figure 5 here, or the contours in their fig. 1). The probable reason lies with the details of the modelling, which generated disc images by generating $N$ particles on eccentric orbits (i.e. as in figures 2 and 3). In that work $N = 10^4$ particles were used, but in the course of this work at least $10^6$ particles were found to be necessary. With small $N$, shot noise renders two model images with identical parameters sufficiently different that their $\chi^2$ can also be very different. This leads to issues with convergence of model fitting, including low acceptance fractions in Markov chain Monte Carlo (MCMC) fitting, and inflated uncertainties. This issue appears to have affected the uncertainties more than the best-fit parameters (which largely agree with the results here), although some differences are discussed below.

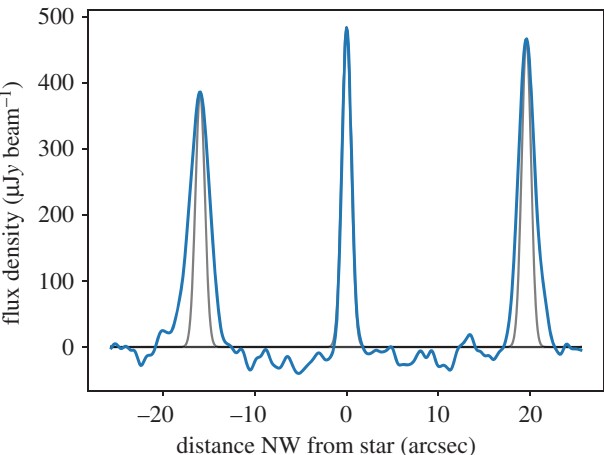

**Figure 5.** Radial surface brightness profile along the disc major axis for Fomalhaut. The blue line shows the data, and the grey lines show Gaussian profiles with FWHM 1.2 arcsec (i.e. the spatial resolution) scaled to each peak. The measured (unconvolved) disc FWHM in each ansa are 2.6 (SE) and 2.1 (NW) arcsec, or 20 and 16 AU.

Finally, MacGregor *et al.* [27] used a uniform range of semi-major axes and in fact yielded the double inner-ring structure discussed above, which can be seen (albeit indistinctly) in their fig. 2. The radial profile in figure 5 does indeed show that the disc is narrower towards the NW ansa (which is near the apocentre), and whether this model provides a good explanation for the data is explored below.

## 3.1. The model

A slightly more complex version of the secular perturbation model used by MacGregor *et al.* [27] was used to model the ALMA data for Fomalhaut and HD 202628. Basically, $N$ particles are generated that sample distributions of eccentricity, inclination, semi-major axis, and true anomaly. These are then binned into a two-dimensional array for the desired viewing geometry to create an image. The fundamental assumption is that the particles have evolved to their current state due to secular perturbations (i.e. are spread evenly around the forced pericentre in $e\omega$-space). This assumption means that the models are created analytically, with no need for $n$-body simulations. From this assumption, and similar secular timescales for inclination evolution, it also follows that the particle nodes are also spread randomly, meaning that any initial misalignment ($i_f$) of the particles with the planet's orbital plane causes the disc to have a height $2i_f$. Viewed edge-on the disc appears brighter at the upper and lower surfaces, because their sinusoidal vertical motions mean that particles spend more time there (i.e. similar to the limb-brightening effect seen in the top right panel of figure 2). While the modelling here does not resolve the scale height of either disc, a finite vertical extent is included to include any effect on the other parameters.

The model parameters are as follows: (i) $x_0$ and $y_0$ allow for any shift of the star+disc position from the observation phase centre, (ii) sky geometry position angle $\Omega$, inclination $i$ and forced pericentre angle $\omega$ (measured from $\Omega$), (iii) total disc and star flux densities $F$ and $F_{\mathrm{star}}$, (iv) disc semi-major axis $a_0$ and Gaussian width $\sigma_a$ or full width $\delta_a$, (v) forced eccentricity $e_f$, proper eccentricity $e_p$ (radius of the circle in $e\omega$-space) and proper eccentricity Gaussian dispersion $\sigma_{e,p}$, (vi) forced inclination $i_f$ and Gaussian inclination dispersion $\sigma_{i,p}$, and (vii) a data reweighting parameter $f_w$ (see below). A total of 15 basic parameters are included, and three more are included in the model for HD 202628 for the position and flux of a bright point source that could otherwise influence the model results. The important difference compared to the model used by MacGregor *et al.* [27] is the inclusion of a proper eccentricity dispersion, which as shown in figure 2 provides an additional means to introduce a finite disc width. Given that the Fomalhaut ring appears narrower at apocentre than pericentre, thus disfavouring a range of semi-major axes, alternative means of generating the disc width are potentially important.

Practically, model images are created with the following steps: (i) the eccentricity parameters $e_p$ and $\sigma_{e,p}$ are used to create $N$ initial particles in a 2-d Gaussian in $e\omega$-space. As these will be precessed into a circle around $e_f$, $e_p$ is taken to be the distance from $e_f$ towards the origin in $e\omega$-space (these are the orange dots in figures 2 and 3). (ii) These particles are then distributed evenly around the forced eccentricity, giving a set of $N$ eccentricities and pericentres (i.e. the blue dots in the same figures), (iii) $N$

inclinations are generated with a Gaussian distribution of dispersion $\sigma_{i,p}$ centred on $i_f$, and these are given randomly chosen ascending nodes, (iv) $N$ random mean anomalies are generated, and using the eccentricity of each particle these are converted into true anomalies, (v) using the eccentricities and true anomalies, the radius of each particle from the star is calculated, and using the true anomaly, pericentre and node, the angular particle locations are calculated. These steps generate an initial three-dimensional particle model of the disc viewed from above, with the forced pericentre measured anticlockwise from the positive $x$-direction. The final steps are (vi) to apply two rotations to move these particles to the observed geometry; first the $y$-coordinates are multiplied by $\cos i$ to incline the disc, and then the $x$, $y$ coordinates are rotated by $\Omega + \pi$ to place the ascending node at the observed position angle, and (vii) finally, the image is generated by binning all particles into a two-dimensional grid, including a $1/\sqrt{r}$ weighting to account for decreasing temperature with radius.

Model images computed using $N = 10^7$ are compared with the ALMA visibilities using the GALARIO [28] software (which returns a $\chi^2$-value for a given model image). Absolute uncertainties are first estimated with the CASA statwt task, which derives weights (=$1/\sigma^2$) based on the variance of the visibilities. The parameter space is explored using MultiNest [29] and emcee [30], with the log likelihood of a given model:

$$\ln \mathcal{L} = -\frac{1}{2}\left(\chi^2 f_w + \sum_i^M 2\ln\frac{2\pi}{w_i f_w}\right), \tag{3.1}$$

where there are $M$ complex visibilities $V$, with weights $w$.[2] While both methods give similar results, the potential advantage of using MultiNest is that it explores *all* parameter space within given ranges, meaning that similarly well-fitting solutions in the parameter space can be identified (though no multi-modal solutions were found here). Only the Multinest results are reported here, which used 75 live points to search a broad parameter space around previously found best-fit parameters. All parameters have uniform priors in linear space.

## 3.2. Fomalhaut

The same seven-pointing data of the Fomalhaut disc that were presented in MacGregor *et al.* [27] were modelled as described above. The data were calibrated using the observatory-supplied script, with the only additional steps being spectral averaging to a single channel per spectral window (spw), 30 s time averaging, and reweighting of all visibilities using the CASA 5.6 statwt task. The signal-to-noise ratio of the disc detection is sufficiently high and the disc sufficiently large that the finite ALMA bandwidth must be accounted for (i.e. the baselines in $uv$ space vary sufficiently across channels that a single frequency cannot be assumed across all channels). The $\chi^2$ are therefore computed using visibilities that assume the average frequency for each spectral window, for each of the seven pointings, and the 28 $\chi^2$-values summed. The precision of the pointings was found to be good enough relative to the resolution that no per-pointing offset parameters were necessary.

Four slightly different models were fitted to the Fomalhaut data; two 'full' models that use all parameters noted above, with 'Gaussian' and 'Uniform' semi-major axis distributions, and two 'simple' models with the same two semi-major axis distributions, and where the eccentricity and inclination dispersion parameters and the forced inclination are set near zero (i.e. vertically flat models that look similar to the upper right or lower left panels in figure 2). From the radial profile in figure 5, it is clear that a purely Gaussian radial distribution is a poor description of the data, and this is confirmed by a poor fit with the Gaussian simple model, so this model is not discussed further. The Gaussian full model, however, yields a very good fit, because the additional radial extent allowed by the eccentricity dispersion can account for surface brightness interior and exterior to the brightest part of the ring. Both Uniform models provide good fits, though the full model does a much better job of reproducing the low surface brightness immediately interior and exterior to the ring. A test using the Schwarz criterion[3] [31] finds that while the second 'penalty' term increases by 45 for the full

---

[2]The factor $f_w$, which the visibility weights are multiplied by, is included because CASA statwt is not guaranteed to produce accurate absolute uncertainties. The log-likelihood function used in the fitting is therefore derived by assuming that the data are distributed as a Gaussian about the model with the correct normalization. The factor two in the summation of equation (3.1) arises because each visibility is assumed to have two independent degrees of freedom (i.e. amplitude and phase).

[3]Also known as the Bayesian information criterion, BIC $= \chi^2 + k\ln n$, where $k$ is the number of parameters and $n$ the degrees of freedom (here $k$ is 12 or 15 for the simple and full models, and $n \approx 1.7 \times 10^6$ for Fomalhaut). Smaller BIC numbers are better, with a decrease of more than about 6 indicating that additional parameters are warranted given the data.

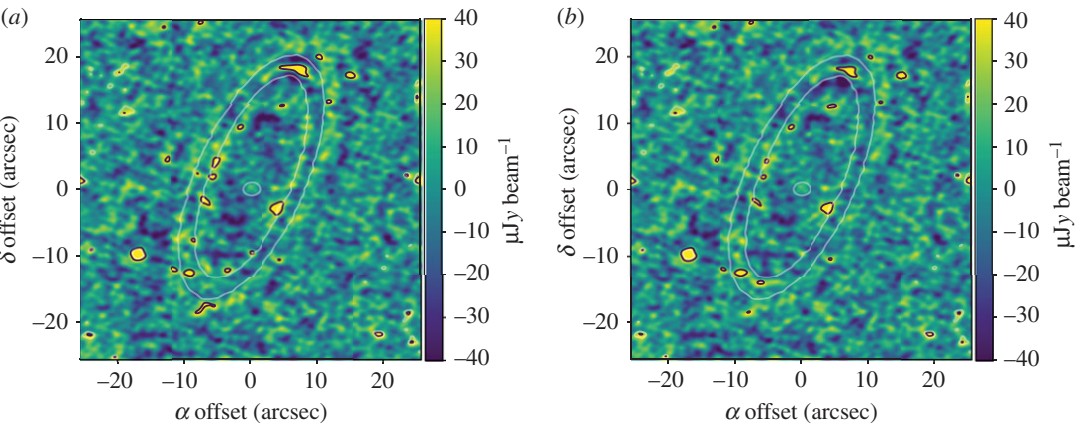

**Figure 6.** Residual maps for the best-fit Uniform simple (*a*) and full (*b*) models of the Fomalhaut ring. The white contour shows the location of the ring, and black contours highlight parts of the image above three times the noise level of 13 μJy beam$^{-1}$.

**Table 1.** Partial list of best-fit model parameters, with the ring radius and width in units of arcseconds and inclinations in radians (full posterior parameter distributions are shown in figures 10–15). Uncertainties are 1$\sigma$, and upper limits are 3$\sigma$.

| | Fomalhaut | | |
|---|---|---|---|
| | Gaussian full | Uniform full | Uniform simple |
| $a_0$ | 18.17 ± 0.04 | 18.18 ± 0.05 | 18.19 ± 0.01 |
| $\sigma_a$ or $\delta_a$ | <0.28 | <1.00 | 2.09 ± 0.06 |
| $e_f$ | 0.125 ± 0.001 | 0.125 ± 0.001 | 0.126 ± 0.001 |
| $e_p$ | 0.019 ± 0.004 | 0.019 ± 0.004 | 0.051 ± 0.001 |
| $\sigma_{e,p}$ | 0.090 ± 0.004 | 0.089 ± 0.002 | — |
| $i_f$ | <0.19 | <0.22 | — |
| $\sigma_{i,p}$ | <0.11 | <0.12 | — |
| | HD 202628 | | |
| | Gaussian full | Uniform full | Uniform simple |
| $a_0$ | 6.6 ± 0.08 | 6.55 ± 0.08 | 6.46 ± 0.04 |
| $\sigma_a$ or $\delta_a$ | <0.4 | <1.2 | <1.3 |
| $e_f$ | 0.12 ± 0.02 | 0.12 ± 0.02 | 0.12 ± 0.01 |
| $e_p$ | <0.08 | <0.08 | <0.08 |
| $\sigma_{e,p}$ | <0.11 | <0.12 | — |
| $i_f$ | <0.5 | <0.5 | — |
| $\sigma_{i,p}$ | <0.4 | <0.4 | — |

models relative to the simple Uniform model, the $\chi^2$ decreases by 240. Such a large decrease indicates that for Fomalhaut the inclusion of these extra parameters is justified by the improvement in the fit.

The modelling results are summarized in table 1 and figure 6, which show the (dirty) residual map for the Uniform models (the Gaussian full model residuals are indistinguishable to the Uniform full model, figures 10–12 show the posterior parameter distributions for most parameters). All models find the same disc position angle of $\Omega = 156.4 \pm 0.1°$ and inclination $i = 66.6 \pm 0.1°$. Figure 6 shows that the models are a good representation of the data, with the only large residual in the NW ansa. This residual noise lies within the ring and has negative residuals interior and exterior, suggesting that the ring is narrower than the model at this location.

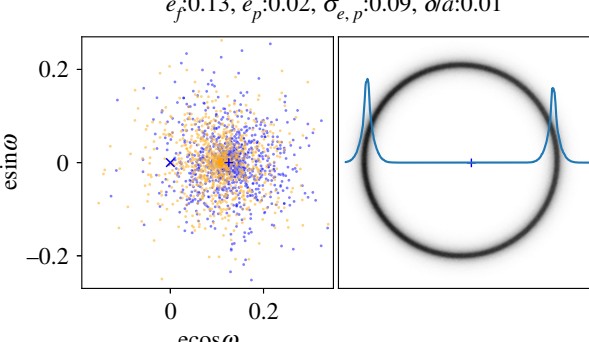

$e_f$:0.13, $e_p$:0.02, $\sigma_{e,p}$:0.09, $\delta/a$:0.01

**Figure 7.** Best-fit Uniform full model to Fomalhaut, shown face-on with pericentre along the positive x-axis. Conventions are as in figure 2. The ring has a narrow bright component whose width is set by $e_p$, and is surrounded by a lower surface brightness 'halo' whose width is set by $\sigma_{e,p}$.

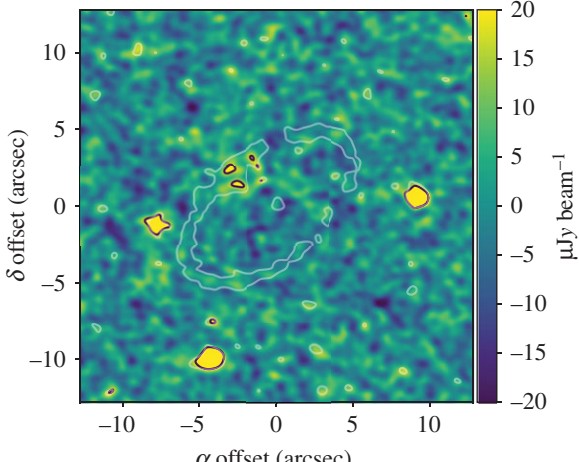

**Figure 8.** Residual map for the best-fit Gaussian full model of the HD 202628 ring. The white contour shows the location of the ring, and black contours highlight parts of the image above three times the noise level of 6 μJy beam$^{-1}$. A bright point source has been subtracted on the NE part of the disc, while three others were not included in the model. All are assumed to be unrelated to the disc.

The Uniform simple model, which is essentially the same as that used by MacGregor *et al.* [27], finds very similar results, but with the expected smaller uncertainties. All models find that the pericentre location is located $41 \pm 1°$ anti-clockwise from the SE disc ansa (i.e. the ascending node). This value is most likely different to the $23 \pm 4°$ found by MacGregor *et al.* [27] for the reasons discussed earlier.

Less obvious from figure 6 is that a greater eccentricity dispersion via the $\sigma_{e,p}$ parameter improves the fit; low level (approx. $1\sigma$) residuals just interior/exterior to the brightest part of the belt remain for the Uniform simple model (*a*), but are reduced for the full models (*b*). The narrower disc width near apocentre means that narrow semi-major axis distributions are preferred, and thus for the full models the disc width is instead generated by a combination of non-zero proper eccentricity and eccentricity dispersion. The best-fit model is shown at high resolution with the $e\omega$-space in figure 7, which shows that the model is composed of a narrow top-hat-like 'core' that is produced by the relatively small typical proper eccentricity $e_p$, and a 'halo' that is produced by the dispersion in proper eccentricity $\sigma_{e,p}$. The semi-major axis distribution does not contribute to the structure, so the Gaussian full model looks the same.

While it was suggested earlier than the Uniform model may be better able to produce a disc that is narrower near apocentre through a quirk of combining a range of semi-major axes and proper eccentricities, this possibility is not borne out by the modelling, primarily because the best models do not require a range of semi-major axes.

Overall, these models show that the disc is consistent with an eccentric ring model, but only if the proper eccentricities are smaller than the forced eccentricity. While the constraints on the model parameters appear very small, this is at least in part because of the restricted nature of the models themselves, which for example are relatively inflexible in terms of semi-major axis distributions. While the conclusion of small proper eccentricities is robust from both empirical and modelling approaches, further insight into the azimuthal dependence of the ring width can only be gained with higher resolution imaging.

## 3.3. HD 202628

The same modelling procedure was applied to HD 202628. The 12m array data from Faramaz *et al.* [25] were obtained from the ALMA archive and calibrated using the observatory-supplied script (the ACA data have insufficient resolution to be valuable here, and are dominated by a background source, so were not included). The CASA `statwt` command was used to reweight the data, and all visibilities were exported and modelled assuming a single wavelength. Compared to Fomalhaut this disc is smaller and detected at lower s/n, so the assumption of a single wavelength was found to be adequate. The same four models were fitted, and all four are able to reproduce the data well and have similar $\chi^2$-values (i.e. including the additional parameters in the full models is not well justified given the data).

The results of the modelling are summarized in table 1 and figure 8, which shows the (dirty) residual map for the Gaussian full model (figures 13–15 show the posterior parameter distributions). All models find the same disc position angle of $\Omega = 130.7 \pm 0.1°$ and inclination $i = 57.4 \pm 0.1°$. The best-fit parameters are in general agreement with Faramaz *et al.* [25], though with a slightly forced larger eccentricity (0.12 here versus their 0.09). Whether the pericentre directions agree is unclear; their fig. 3 suggests that pericentre lies near the NW ansa, but figure 10 puts it more directly west of the stellar position. The latter appears consistent with the value of 143° from the SE ansa found here so seems more likely to be correct. These differences may be related to the difference in eccentricity, and will be addressed in the near future [32].

As with Fomalhaut, the residuals show that the model is a good representation of the data, though achieving this is less of a challenge for HD 202628 given the much lower s/n. No significant residuals that are clearly related to the disc remain, though imperfect subtraction of the bright point source NW of the star is apparent. While the eccentricity and disc width parameters $\sigma_a$ and $e_p$ are given as upper limits, this is based on their one-dimensional posterior distributions; inspection of the two-dimensional distributions finds that these two parameters cannot simultaneously be zero, implying that the ring is radially resolved (albeit marginally).

As with Fomalhaut, the best fit model finds that the proper eccentricity is significantly lower than the forced eccentricity,[4] again implying that the ring cannot be modelled as a secularly perturbed ring whose eccentricities were initially near zero. The lower s/n and spatial resolution means that the constraints on the parameters are not as strong, and higher spatial resolution data would be needed to obtain any insight into how the ring width varies as a function of azimuth.

## 4. Eccentric ring origins: why so narrow?

Having established that the Fomalhaut and HD 202628 debris rings are too narrow to be explained by secularly perturbed particles that started with near-zero initial eccentricities, several possible origins are discussed. Figure 9 is shown to frame this discussion, which shows the eccentricity and relative width of selected debris discs (see [36] for an extensive list). Only discs with ALMA-measured widths and eccentricities are shown, since these provide a quantity that is not affected by small dust grain dynamics (the eccentricity for HR 4796 was not detected by ALMA, so relies on scattered light observation, but is included because the structure at both wavelengths is consistent). While other discs have also been observed with ALMA, these are wider than those shown, and are not known to have significant eccentricity. The possible importance of figure 9 is that the narrower discs are also those that are most eccentric. Fomalhaut and HD 202628 both lie above the dashed line, indicating that their width is narrower than expected given their forced eccentricities. HR 4796 is a marginal case—

---

[4]This significance is better verified directly from the posterior parameter distributions in appendix A, as the values in table 1 do not reflect asymmetric uncertainty ranges, nor that the 3$\sigma$ limit for a parameter is not necessarily three times the 1$\sigma$ uncertainty.

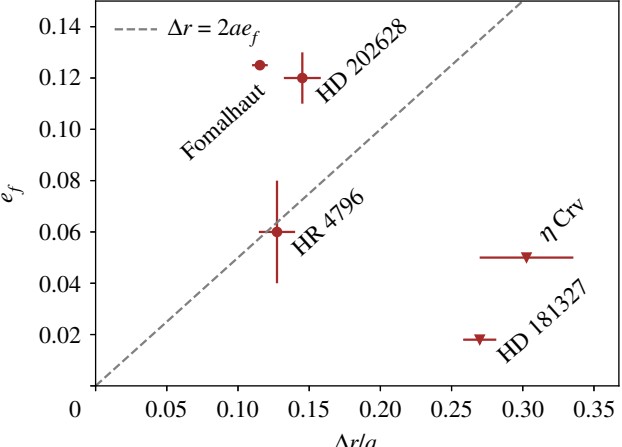

**Figure 9.** Forced eccentricity and relative ring width of selected debris discs. Widths are taken from modelling results [25,33–35], with the exception of Fomalhaut, which uses the FWHM of the narrower ansa (figure 5). The line shows the expected width of the belt in a zero initial eccentricity secular perturbation scenario, showing that Fomalhaut and HD 202628's discs are narrower than expected, while HR 4796 is marginal (and like HD 202628 is not well resolved radially, so may be narrower). $\eta$ Corvi and HD 181327's discs are consistent with being circular, so their width originates from a range of semi-major axes, or eccentric bodies with randomly distributed pericentre directions.

Kennedy *et al.* [35] note that the disc is only marginally resolved, and could be narrower (i.e. further to the left of the dashed line).

Empirically, an important goal seems to be to fill out figure 9, in particular for discs whose widths are ≲30% of their semi-major axis; if the trend holds then it may be reasonably expected that the mechanism that radially concentrates planetesimals also excites a forced eccentricity. Figure 9 may eventually reveal a demarcation between narrow/eccentric and wide/circular disc systems, which could indicate whether the concentration mechanism operated in a given system.

The possible origins of narrow eccentric debris rings may be grouped into two main types of scenarios, depending on whether the approximately millimetre-sized dust observed by ALMA does, or does not, trace the planetesimal population. In the former case the narrow rings are explained by either modifying the secular perturbation scenario to incorporate non-zero initial eccentricities, or perhaps discarding it entirely, while in the latter case secular perturbations may be retained, but the orbits of bodies change as a function of their size (i.e. particle free eccentricities 'damp' towards the forced eccentricity as they move down the collisional cascade). These scenarios, plus several others, are discussed below.

## 4.1. Eccentric initial conditions

The simplest explanation for the narrow debris rings is that the (perhaps naive) expectation of small initial eccentricities in the secular perturbation scenario is not met. That is, the initial conditions in $e\omega$-space may look more like the left two panels in figure 2 than the upper right panel, though the initial eccentricities need not be at the location of the forced eccentricity. In this case there are two main scenarios to consider.

One possibility is that the planetesimals acquire significant coherent eccentricities (i.e. with a preferred pericentre direction) prior to the evolution that is assumed in a standard secular perturbation scenario. For example, a single planet may exert some influence on planetesimals prior to dispersal of the gas disc, but this influence is diminished in a way that simply decreases the magnitude of the forced eccentricity (but with approximately the same forced pericentre angle as after dispersal, perhaps similar to the scenario discussed by Rafikov [37] for circumbinary planetesimals). Thus, when the gas is removed the planetesimal eccentricities are already shifted somewhat towards the forced eccentricity, and as they precess under the influence of the same planet, they trace out a smaller circle in $e\omega$-space, resulting in a narrower debris ring. To also provide the tentative trend in figure 9, this process should also radially concentrate planetesimals, perhaps by trapping them in a gas pressure maximum exterior to the planet

(e.g. [38]), or trapping dust in a narrow ring which then goes on to form said planetesimals via the streaming instability (e.g. [39]).

It is of course possible that the planetesimals acquire *all* of their eccentricity before the gas disc is dispersed, in which case no further eccentricity evolution via secular perturbations is necessary. In this case, there is not necessarily a need for any planet, as it may be that the initial eccentricity excitation is related to planet-free gas dynamics or dust/gas interaction, For example, Lyra & Kuchner [40] show that narrow eccentric rings could form in planet-free discs with high dust/gas ratios, which could occur in local gas pressure maxima, or more globally as the density drops during gas disc dispersal.

How might evidence for these possible scenarios be sought? Most simply, they both predict that planetesimals should form and acquire non-zero eccentricities during the gas-rich phase of protoplanetary disc evolution. One system showing a possible asymmetry is PDS 70, which hosts a protoplanetary disc and at least one interior planet [41,42]. The dust component is fairly narrow and shows both a brightness asymmetry (the NW ansa is brighter [43]), and a stellocentric offset that places this ansa farther from the star (e.g. visible in fig. 2 of [44]). While these combined properties might be interpreted as apocentre glow (e.g. [45]), the disc is probably optically thick and the origin of the asymmetry may not be so simple. Nevertheless, PDS 70 is a possible precursor to systems such as Fomalhaut and HD 202628, and thus circumstantial evidence that eccentric rings can exist before the debris phase.

A possible difference between the single and no-planet scenarios is that in the presence of an eccentric planet the eccentricity of debris rings should increase with time. Immediately after gas disc dispersal, secular perturbations have yet to act to their full effect, so debris rings only possess their initial eccentricity, but later acquire their full eccentricity, which is larger. A young disc such as that around HD 4796, with eccentricity ≈0.06 (i.e. less than the much older systems Fomalhaut and HD 202628, [46]) is suggestive, but there are too few discs with well-constrained eccentricities and widths for this test to be possible at present.

## 4.2. Eccentricity damping

While the prior discussion assumed that the observed millimetre-wave disc structure is a true reflection of the underlying planetesimal orbits, this assumption might be violated. Specifically, if particle free eccentricities decrease as they become smaller, a debris ring could appear narrower at millimetre wavelengths than the underlying planetesimal belt. A possible way to damp orbital eccentricities in a debris disc is in collisions, if the random velocities of post-collision fragments tend to be smaller than the targets'. Because this damping decreases the relative orbital velocities, the effect in an eccentric disc with a preferred pericentre direction is to drive down the free eccentricity. This scenario was explored analytically by Pan & Schlichting [47], and in general finds the expected decrease in velocities for smaller objects.

This scenario, however, essentially relies on destructive collisions being between objects of similar size, as the orbit of the centre of mass has a lower random velocity than either of the two bodies. If the typical destructor is much smaller than the target body, then the post-collision fragments will tend to retain their original velocities, and the damping is inefficient.[5] A significant size difference is generally expected for destructive collisions, because the size distribution is always such that smaller objects are much more common, and therefore the impactor that destroys a larger object is usually the smallest one that can do so [48]. In this case, orbital velocities do not decrease significantly with object size, and the entire size distribution inherits eccentricities from the largest bodies.

This picture lacks some nuance, and for example ignores the effect of non-destructive collisions, but collisional damping seems unlikely to be significant, and if anything eccentricities seem more likely to increase in collisions. For example, in their simulations of impacts with 100 km diameter targets, Jutzi *et al.* [49] find that the largest fragments, which dominate the mass, tend to have low velocities relative to the target centre of mass, so the approximation that all bodies share the same orbit as the collision centre of mass (e.g. [50]) is reasonable.

A potentially attractive aspect for collisional damping is that it could be observationally testable. If free eccentricities tend to decrease with particle size in the mm to cm size regime, observations may

---

[5]Picture two people throwing pumpkins towards each other at high velocity; neither is likely to be splattered with fragments (most will fall to the ground below the collision). However, if one throws a much smaller object, such as a pebble, the pebble thrower will be hit by the fragmented pumpkin (which has much more momentum).

be able to measure different debris ring widths as a function of wavelength. While the discussion above suggests that significant damping is unlikely, modelling and observations focusing on this specific size range would be valuable.

A side effect of damping is that the relative velocities between smaller particles are also smaller, leading to longer collisional lifetimes and a steeper size distribution than would be expected if all objects share the same random velocities [47]. The steeper size distribution in turn means that for a given observed disc brightness (which is proportional to the surface area of small grains), the inferred total mass (which is dominated by large bodies) can be much smaller. Damping therefore provides a possible resolution of the debris disc 'mass problem' (see [51] for discussion of this issue), in which disc masses in systems such as HR 4796 are inferred to be implausibly large (e.g. [35]).

## 4.3. A single event

A third way to explain the narrow debris rings is to discard the standard debris disc paradigm entirely (i.e. the idea of dust derived from an underlying population of planetesimals undergoing continuous collisions in a pseudo-steady state). That is, to consider transient and/or stochastic events as a way to produce a population of objects on similar eccentric orbits. The scenario for creating a narrow debris ring is then based on the breakup of a single large body, which must have been destroyed in such a way that the velocity dispersion of fragments is small (i.e. similar to described above). The scenario may therefore be very similar to that explored by Jackson *et al.* [52], where debris is released from a large body in a collision. This scenario tends to produce non-axisymmetric dust density distributions, as more dust is concentrated at the spatial location of the original dust production event (and more radially distributed on the opposite side of the star). However, for a small velocity dispersion the asymmetry will be smaller, and if there are other perturbing bodies in the system, the asymmetry is decreased as the fragments' orbits precess.

In some ways this scenario is similar to a normal debris ring scenario, in that once created, a population of fragments evolves in the same way, and is indistinguishable from a planetesimal population with similar orbits. It is therefore very hard to rule out that narrow debris rings (eccentric or otherwise) originate from planetesimals that are themselves a family of fragments.[6] A basic objection, however, is why only a single debris ring should be detected around a given star, and not a series of near-concentric rings that arise from similar events. A more serious issue is that inferred masses for debris discs can be of the order of tens of Earth masses (e.g. 20–30$M_\oplus$ for Fomalhaut, [48]), and it therefore seems unlikely that this mass in solid fragments can be created from a single event.

## 4.4. Shepherding planets

Boley *et al.* [54] proposed that the width of the Fomalhaut debris ring is constrained by a pair of shepherding planets, by analogy with Uranus' $\varepsilon$ ring and Saturn's F ring. In these cases the moons were inferred to exist prior to their discovery because the rings should rapidly spread radially, and some external force is required to confine them [55]. As discussed above, collisions in debris discs are such that significant spreading of narrow rings is not necessarily expected. This picture does not, however, mean that shepherding moons do not exist, just that they are not required by the analogy that narrow debris rings appear similar to narrow planetary rings.

As discussed at the outset, a problem with planet-interaction scenarios is that the putative planets are often not detectable. In the case of Fomalhaut, the inferred shepherding planet masses were several Earth masses [54], meaning that their detection at approximately 140 AU from the star is currently impossible. An avenue that has not yet been explored is whether shepherding planets should induce detectable azimuthal structure in the ring edges near the planets [56]; such a study would require simulations, and no doubt higher spatial resolution imaging. In the meantime, the shepherding planet scenario is considered more complicated than is required by the data; while shepherding requires two eccentric planets, the secular perturbation scenario only requires one.

---

[6]Another difference noted by Cataldi *et al.* [53] is that if CO gas is also liberated by the collisional evolution, then very different levels of atomic carbon (which results from rapid photodissociation of CO) may result. As carbon is not necessarily easily removed from the system, it builds up over time, so a system where the observed debris is the result of a more recent event should have lower levels of carbon than one that has been evolving for longer.

## 4.5. Massive debris ring

The secular perturbation theory as applied to debris discs usually assumes that the belt mass can be ignored; the planet perturbs the belt, but the belt does not perturb the planet (i.e. cause it to precess). In the case of eccentric debris rings this assumption seems justified, as a precessing planet would cause the forced eccentricity to move in $e\omega$-space over time, and differing precession rates for ring particles at different semi-major axes would cause the ring to quickly lose the observed coherence. However, this issue might be circumvented with shepherding satellites as discussed above, or if the ring has sufficient mass that self-gravity forces all particle pericentres to remain similar. If such a coherence-maintaining mechanism could operate, then it is possible that the debris ring was initially near-circular, and gained an observable eccentricity through mutual interaction with an initially eccentric planet. While these ideas were developed for planetary rings in the Solar system (e.g. [55]), whether they apply here is not clear, so this scenario requires further work.

## 5. Conclusion

This paper shows that the eccentric Fomalhaut and HD 202628 debris rings are narrower than expected, based on a secular perturbation model and the expectation of near-zero initial eccentricities. In the case of Fomalhaut, this narrowness is clear from simply measuring the radial profile as observed by ALMA (figure 5), but for HD 202628 the s/n per beam is lower, so the conclusion is drawn from several similar eccentric ring models.

What does this narrowness mean? The most likely implication is that in a secular perturbation scenario the planetesimals did not initially have near-circular orbits. This prior excitation could be a consequence of planetary perturbations within the gas-rich protoplanetary disc, but if other processes can produce coherently eccentric planetesimal orbits, may not require a planet at all. In either case, however, a clear prediction is that eccentric rings should exist within protoplanetary discs. Whether these rings are observable is less clear, but PDS 70, host to a protoplanetary disc that shows both geometric and brightness asymmetry, and at least one planet, is singled out as a possible progenitor of systems such as Fomalhaut and HD 202628. Making and extending such links would be valuable to understand whether debris discs or their direct progenitors exist within gas-rich protoplanetary discs.

An alternative possibility is that the planetesimal belt is wider than observed with ALMA, but that objects are damped as they fragment to smaller sizes. This possibility is disfavoured, but may be testable if millimetre- to centimetre-size grains have different enough eccentricities. Observational tests of collisional damping may also be relevant to the debris disc mass problem.

A plot of belt-forced eccentricity against relative radial width (figure 9) provides circumstantial evidence of a trend that the narrowest debris rings are also the most eccentric. However, a lack of eccentric systems limits the numbers in this figure, and more systems will need to be characterized to explore this possible relation further. For a start, HR 4796 is identified as possibly being narrower than expected, and should be imaged at higher spatial resolution.

Data accessibility. This paper makes use of the following ALMA data: ADS/JAO.ALMA#2015.1.00966.S, ADS/JAO.ALMA#2016.1.00515.S, which are available from the ALMA archive: http://almascience.nrao.edu/aq/. Relevant code for this research work is stored in GitHub: https://github.com/drgmk/eccentric-width and has been archived within the Zenodo repository: https://doi.org/10.5281/zenodo.3832148.

Competing interests. I declare I have no competing interests.

Funding. G.M.K. is supported by the Royal Society as a Royal Society University Research Fellow.

Acknowledgements. Thanks to both referees for constructive reports, Jane Huang for noting the possibly eccentric structure of the PDS 70 disc, to Luca Matrà for advice/discussions on modelling ALMA data, to Meredith MacGregor and Virginie Faramaz for discussions on their prior work on Fomalhaut and HD 202628, and to Mark Wyatt for the reminder that planetesimal belts have mass. ALMA is a partnership of ESO (representing its member states), NSF (USA) and NINS (Japan), together with NRC (Canada), MOST and ASIAA (Taiwan), and KASI (Republic of Korea), in cooperation with the Republic of Chile. The Joint ALMA Observatory is operated by ESO, AUI/NRAO and NAOJ.

## Appendix A. Posterior distributions for model fitting

In figures 10–15.

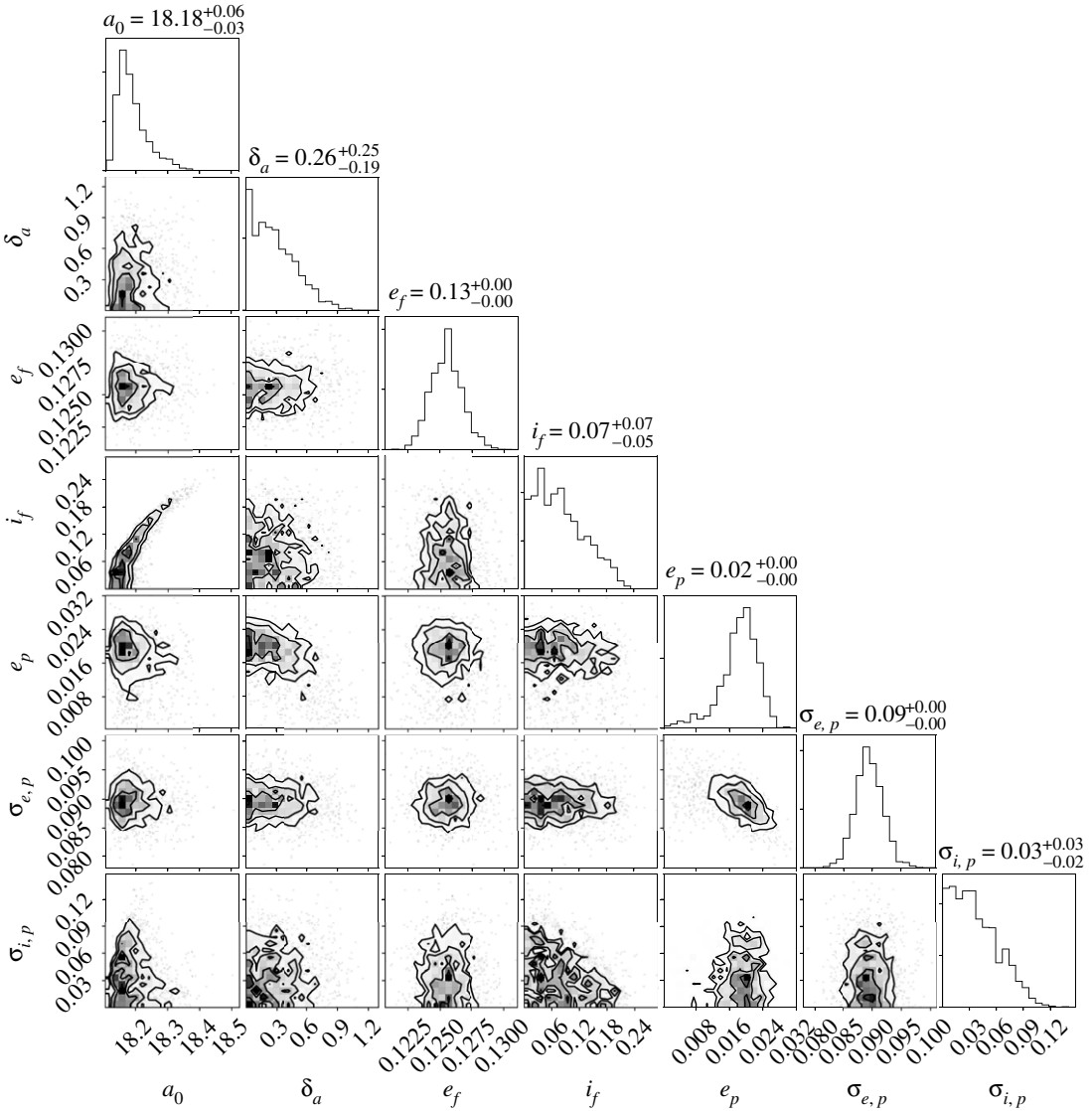

**Figure 10.** Posterior parameter distributions for the full model of the Fomalhaut ring with a uniform distribution of semi-major axes. Only parameters related to the eccentric disc model are shown here. Each off-diagonal panel shows the two-dimensional distributions, with 1, 2 and 3$\sigma$ contours in regions of high density, and individual dots in regions of low density. The diagonal panels show the one-dimensional distributions.

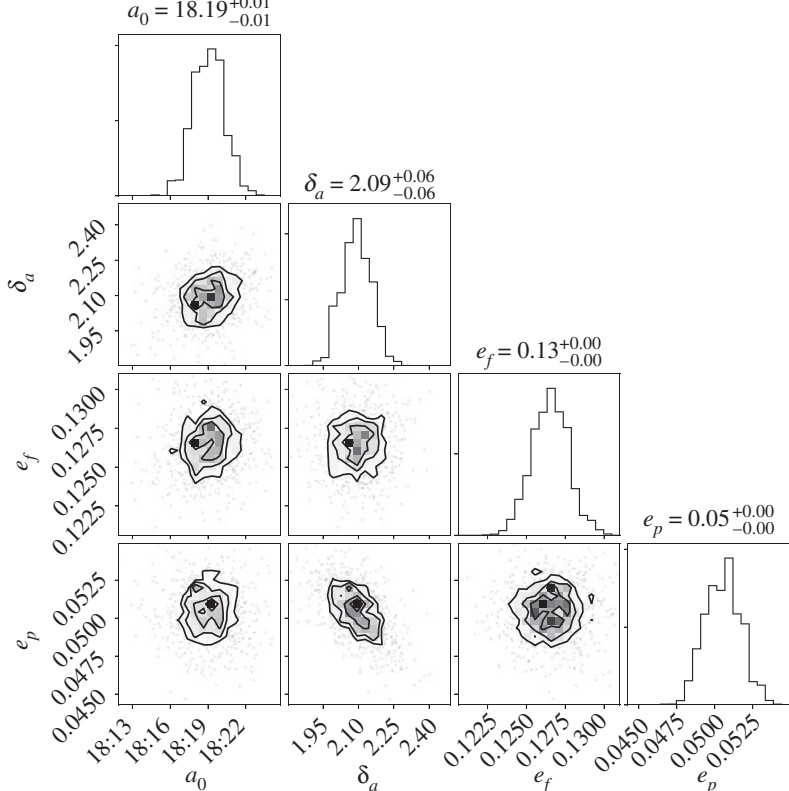

**Figure 11.** Posterior parameter distributions for the simple model of the Fomalhaut ring with a uniform range of semi-major axes. Only parameters related to the eccentric disc model are shown here. Each off-diagonal panel shows the two-dimensional distributions, with 1, 2 and $3\sigma$ contours in regions of high density, and individual dots in regions of low density. The diagonal panels show the one-dimensional distributions.

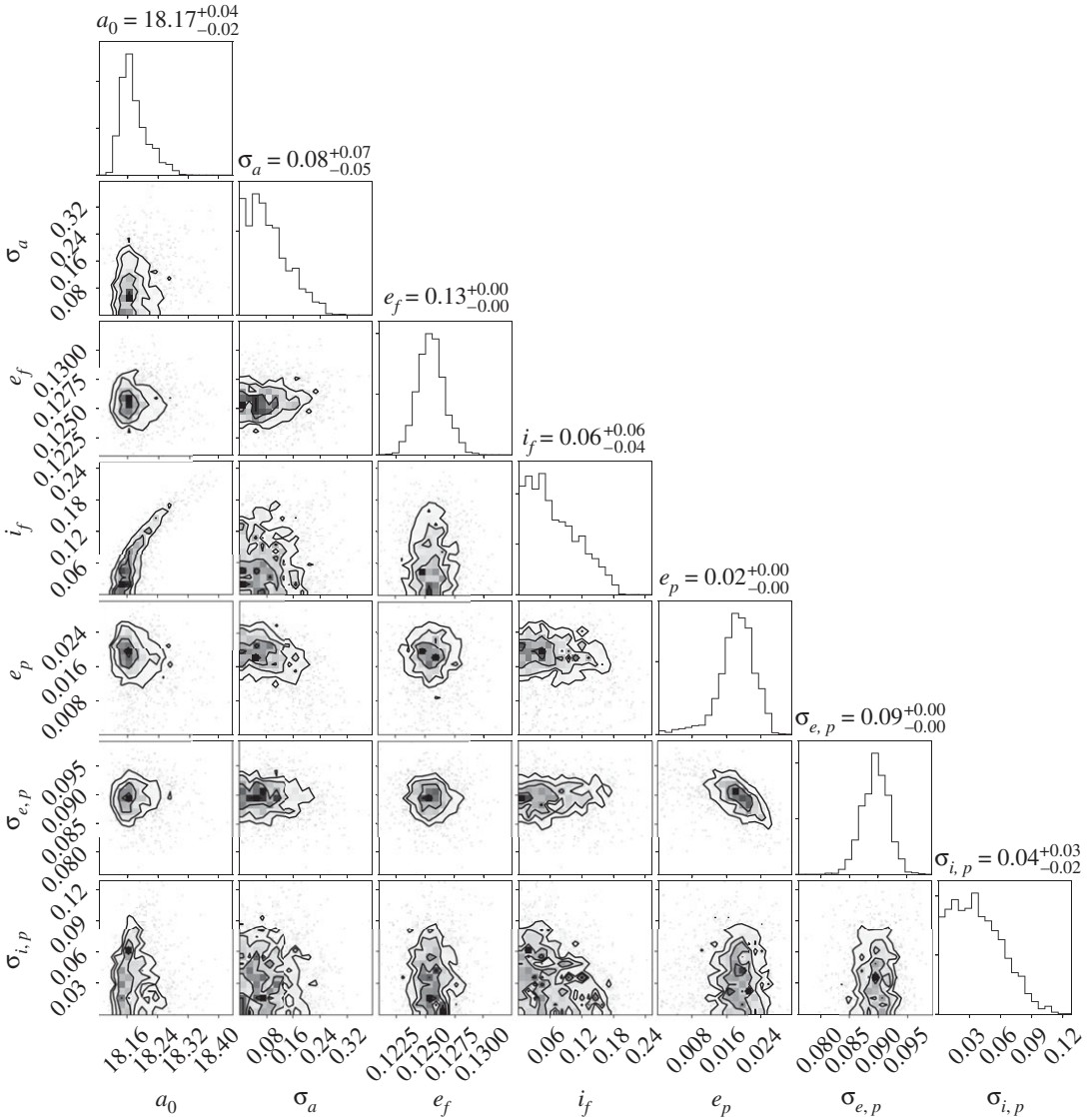

**Figure 12.** Posterior parameter distributions for the full model of the Fomalhaut ring with a Gaussian range of semi-major axes. Only parameters related to the eccentric disc model are shown here. Each off-diagonal panel shows the two-dimensional distributions, with 1, 2 and $3\sigma$ contours in regions of high density, and individual dots in regions of low density. The diagonal panels show the one-dimensional distributions.

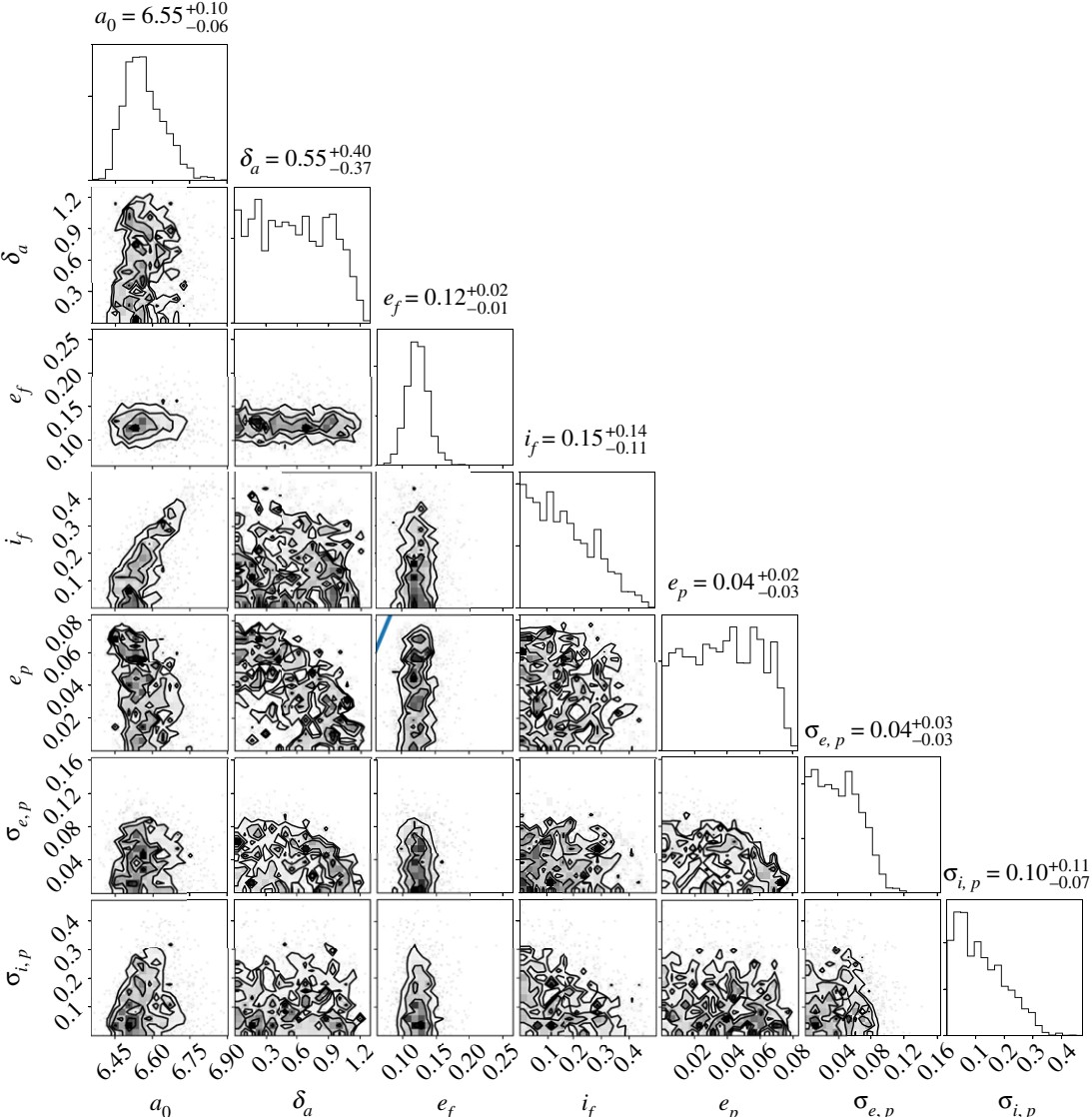

**Figure 13.** Posterior parameter distributions for the full model of the HD 202628 ring with a uniform range of semi-major axes. A line with $e_p = e_f$ is shown in the appropriate panel, illustrating that the proper eccentricity is significantly lower than the forced eccentricity. Only parameters related to the eccentric disc model are shown here. Each off-diagonal panel shows the two-dimensional distributions, with 1, 2 and $3\sigma$ contours in regions of high density, and individual dots in regions of low density. The diagonal panels show the one-dimensional distributions.

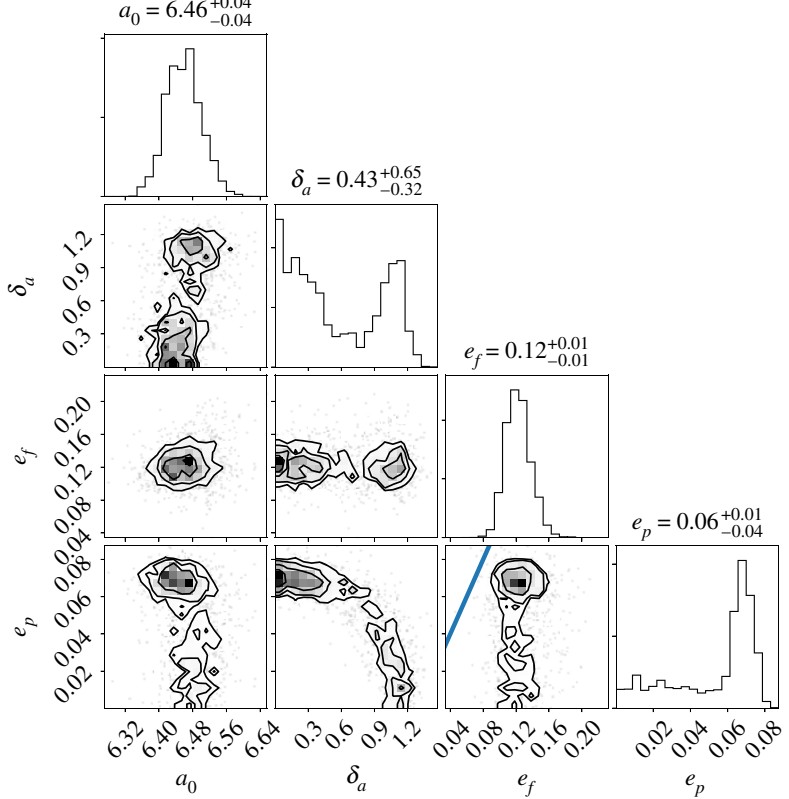

**Figure 14.** Posterior parameter distributions for the simple model of the HD 202628 ring with a uniform range of semi-major axes. A line with $e_p = e_f$ is shown in the appropriate panel, illustrating that the proper eccentricity is significantly lower than the forced eccentricity. Only parameters related to the eccentric disc model are shown here. Each off-diagonal panel shows the two-dimensional distributions, with 1, 2 and 3$\sigma$ contours in regions of high density, and individual dots in regions of low density. The diagonal panels show the one-dimensional distributions.

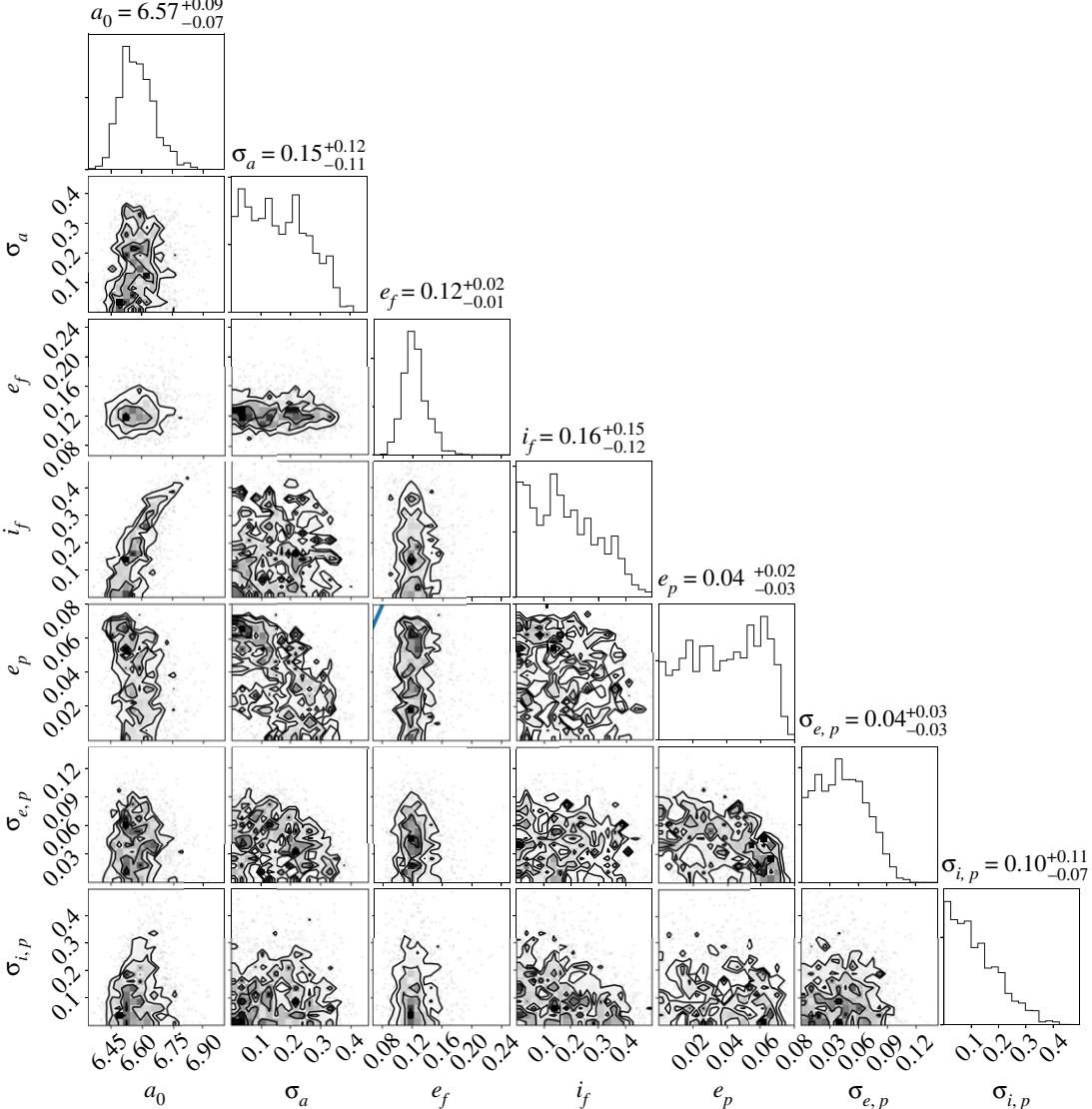

**Figure 15.** Posterior parameter distributions for the full model of the HD 202628 ring with a Gaussian range of semi-major axes. A line with $e_p = e_f$ is shown in the appropriate panel, illustrating that the proper eccentricity is significantly lower than the forced eccentricity. Only parameters related to the eccentric disc model are shown here. Each off-diagonal panel shows the two-dimensional distributions, with 1, 2 and $3\sigma$ contours in regions of high density, and individual dots in regions of low density. The diagonal panels show the one-dimensional distributions.

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
