## [Reviewer comments · Royal Society Open Science]

Review History

RSOS-200063.R0 (Original submission)

Review form: Reviewer 1

Is the manuscript scientifically sound in its present form?

Yes

Are the interpretations and conclusions justified by the results?

Yes

Is the language acceptable?

Yes

Do you have any ethical concerns with this paper?

No

Have you any concerns about statistical analyses in this paper?

No

Recommendation?

Accept with minor revision (please list in comments)

Comments to the Author(s)

This manuscript presents an interesting study on the radial width and eccentricity of a small sample of debris disks recently observed by ALMA.

In particular, it focuses on the geometry of two well-studied systems, Fomalhaut and HD 202628, which show very narrow and eccentric rings.

After deriving new constraints on the geometry of those two debris disks via model-fitting of the ALMA observational data, the author shows how these characteristics cannot be reconciled with the often-assumed scenario of secularly perturbed planetesimals that started with zero (or near-zero) initial eccentricities.

Possible alternative scenarios that may explain the observed geometries are discussed, together with some ideas for future observations that may test the predictions from those scenarios.

Although the available data does not allow the author to obtain conclusive results on the origin of these narrow and eccentric debris disks, the analysis presented here will be certainly useful for future similar investigations on the dynamics of these systems.

For these reasons, I suggest to have this manuscript published after the authors address my comments listed below.

- My main comments refer to the presentation of the models used by the author. More details should be provided both for the procedure adopted to run the N-body simulations discussed throughout the text, as well as on the method used to convert the results of the N-body simulations into images for comparison with the ALMA interferometric data.

- Fig. 1: I would suggest to give more explicit definitions of all the parameters shown in Figure 1, for example inside the figure caption.

- When describing Fig. 1 in lines 10-12 of pag. 5, it is not immediately clear what the author means by "range of eccentricities" vs "dispersion of eccentricities".

- Pag. 5, lines 45-47) It would be important to spell out the specific "reasonable" assumptions about the initial conditions.

- When discussing the ALMA data for both Fomalhaut and HD 202628, it would be important to at least state some of the key properties of those observations, e.g. wavelength and angular resolution.

- Fig. 8: it would be useful to add errorbars, when possible, to the datapoints shown on that plot.

- Pag. 13, line 43: "indicating that their width is greater": does the author mean "lower" instead?

- Fig. 9-14: I would suggest to increase the size of the characters, right now they are very difficult to read in the printed version.

- Typos:

Pag. 12, line 44: "acheiving"

Pag. 12, line 52: "resoution"

Pag. 16, line 18: "pair shepherding planets"

Review form: Reviewer 2

Is the manuscript scientifically sound in its present form?

Yes

Are the interpretations and conclusions justified by the results?

Yes

Is the language acceptable?

Yes

Do you have any ethical concerns with this paper?

No

Have you any concerns about statistical analyses in this paper?

No

Recommendation?

Accept with minor revision (please list in comments)

Comments to the Author(s)

This manuscript describes a re-analysis of ALMA data on the Fomalhaut and HD 202628 debris rings with special attention to the ring's radial widths and their constraints on the orbits of the planetesimals/dust making up the rings; the results suggest that in each ring, the proper and forced eccentricities of the dust particles differ significantly. I recommend publication after some relatively minor points (listed below) are addressed.

1) The discussion of the Fomalhaut analysis notes that the "full" models improve the fit over the "uniform simple" model. While it's explained in the text that the inclusion of the proper eccentricity dispersion specifically helps to capture the disc radial profile, it could also be said that adding parameters to the fitted model should generally improve the fit. Is there a way to quantify the fit improvement and show that it's better than what would be expected simply from adding a parameter?

2) I think the extra discussion of Fig 1 is a good idea given the use of similar figures later, but I think the presentation could be clearer, especially as it seems to be targeted toward readers who aren't very familiar with the dynamics. Eg, it isn't immediately clear from the text what the difference between "range" in eccentricity and eccentricity "dispersion" is (later in the text, the two terms are used somewhat interchangeably), or why two variables are needed to represent the "random component" of the eccentricity. Perhaps the dynamics could be explained before describing the figure's different examples of this dynamics -- that way all the relevant variables are introduced completely and the reader has the tools to see what is going on in the figure when it's brought up for the first time.

3) In section 4 and the later parts of section 3 it became clear that the usual expectation was that the forced and free eccentricities should be roughly equal, but I couldn't find an explanation/discussion of that expectation in the text.

4) The discussion of the prior Fomalhaut analysis motivates the current reanalysis pretty clearly, but it was less clear from the description of the HD 202628 prior work why a reanalysis was called for. While a comparison with the Faramaz 2019 paper shows that the modelling goals and procedures are rather different in the present work, it would be nice to add some discussion of the differences to the text in the beginning of section 3.

5) Finally, it would be great if the size of the text in the figure labels could be larger, to make the subscripts and appendix figures especially easier to read.

Decision letter (RSOS-200063.R0)

03-Apr-2020

Dear Dr Kennedy

First, I would like to apologise for the delay in getting back to you.

On behalf of the Editors, I am pleased to inform you that your Manuscript RSOS-200063 entitled "The unexpected narrowness of eccentric debris rings: a sign of eccentricity during the protoplanetary disc phase" has been accepted for publication in Royal Society Open Science subject to minor revision in accordance with the referee suggestions. Please find the referees' comments at the end of this email.

The reviewers have recommended publication, but also suggest some minor revisions to your manuscript. Therefore, I invite you to respond to the comments and revise your manuscript.

- Ethics statement

- Data accessibility

If you wish to submit your supporting data or code to Dryad (<http://datadryad.org/>), or modify your current submission to dryad, please use the following link:
<http://datadryad.org/submit?journalID=RSOS&manu=RSOS-200063>

- Competing interests

- Authors' contributions

AB carried out the molecular lab work, participated in data analysis, carried out sequence alignments, participated in the design of the study and drafted the manuscript; CD carried out

the statistical analyses; EF collected field data; GH conceived of the study, designed the study, coordinated the study and helped draft the manuscript. All authors gave final approval for publication.

- Acknowledgements

- Funding statement

Because the schedule for publication is very tight, it is a condition of publication that you submit the revised version of your manuscript before 12-Apr-2020. Please note that the revision deadline will expire at 00.00am on this date. If you do not think you will be able to meet this date please let me know immediately.

If your manuscript is newly submitted and subsequently accepted for publication, you will be asked to pay the article processing charge, unless you request a waiver and this is approved by Royal Society Publishing. You can find out more about the charges at <https://royalsocietypublishing.org/rsos/charges>. Should you have any queries, please contact openscience@royalsociety.org.

on behalf of Rob Ivison (Subject Editor)
openscience@royalsociety.org

Reviewer comments to Author:
Reviewer: 1

Comments to the Author(s)

This manuscript presents an interesting study on the radial width and eccentricity of a small sample of debris disks recently observed by ALMA.

In particular, it focuses on the geometry of two well-studied systems, Fomalhaut and HD 202628, which show very narrow and eccentric rings.

After deriving new constraints on the geometry of those two debris disks via model-fitting of the ALMA observational data, the author shows how these characteristics cannot be reconciled with the often-assumed scenario of secularly perturbed planetesimals that started with zero (or near-zero) initial eccentricities.

Possible alternative scenarios that may explain the observed geometries are discussed, together with some ideas for future observations that may test the predictions from those scenarios.

Although the available data does not allow the author to obtain conclusive results on the origin of these narrow and eccentric debris disks, the analysis presented here will be certainly useful for future similar investigations on the dynamics of these systems.

For these reasons, I suggest to have this manuscript published after the authors address my comments listed below.

- My main comments refer to the presentation of the models used by the author. More details should be provided both for the procedure adopted to run the N-body simulations discussed

throughout the text, as well as on the method used to convert the results of the N-body simulations into images for comparison with the ALMA interferometric data.

- Fig. 1: I would suggest to give more explicit definitions of all the parameters shown in Figure 1, for example inside the figure caption.

- When describing Fig. 1 in lines 10-12 of pag. 5, it is not immediately clear what the author means by "range of eccentricities" vs "dispersion of eccentricities".

- Pag. 5, lines 45-47) It would be important to spell out the specific "reasonable" assumptions about the initial conditions.

- When discussing the ALMA data for both Fomalhaut and HD 202628, it would be important to at least state some of the key properties of those observations, e.g. wavelength and angular resolution.

- Fig. 8: it would be useful to add errorbars, when possible, to the datapoints shown on that plot.

- Pag. 13, line 43: "indicating that their width is greater": does the author mean "lower" instead?

- Fig. 9-14: I would suggest to increase the size of the characters, right now they are very difficult to read in the printed version.

- Typos:

Pag. 12, line 44: "acheiving"

Pag. 12, line 52: "resoution"

Pag. 16, line 18: "pair shepherding planets"

Reviewer: 2

Comments to the Author(s)

This manuscript describes a re-analysis of ALMA data on the Fomalhaut and HD 202628 debris rings with special attention to the ring's radial widths and their constraints on the orbits of the planetesimals/dust making up the rings; the results suggest that in each ring, the proper and forced eccentricities of the dust particles differ significantly. I recommend publication after some relatively minor points (listed below) are addressed.

1) The discussion of the Fomalhaut analysis notes that the "full" models improve the fit over the "uniform simple" model. While it's explained in the text that the inclusion of the proper eccentricity dispersion specifically helps to capture the disc radial profile, it could also be said that adding parameters to the fitted model should generally improve the fit. Is there a way to quantify the fit improvement and show that it's better than what would be expected simply from adding a parameter?

2) I think the extra discussion of Fig 1 is a good idea given the use of similar figures later, but I think the presentation could be clearer, especially as it seems to be targeted toward readers who aren't very familiar with the dynamics. Eg, it isn't immediately clear from the text what the difference between "range" in eccentricity and eccentricity "dispersion" is (later in the text, the two terms are used somewhat interchangeably), or why two variables are needed to represent the "random component" of the eccentricity. Perhaps the dynamics could be explained before describing the figure's different examples of this dynamics -- that way all the relevant variables are introduced completely and the reader has the tools to see what is going on in the figure when it's brought up for the first time.

3) In section 4 and the later parts of section 3 it became clear that the usual expectation was that

the forced and free eccentricities should be roughly equal, but I couldn't find an explanation/discussion of that expectation in the text.

4) The discussion of the prior Fomalhaut analysis motivates the current reanalysis pretty clearly, but it was less clear from the description of the HD 202628 prior work why a reanalysis was called for. While a comparison with the Faramaz 2019 paper shows that the modelling goals and procedures are rather different in the present work, it would be nice to add some discussion of the differences to the text in the beginning of section 3.

5) Finally, it would be great if the size of the text in the figure labels could be larger, to make the subscripts and appendix figures especially easier to read.

Author's Response to Decision Letter for (RSOS-200063.R0)

See Appendix A.

Decision letter (RSOS-200063.R1)

Dear Dr Kennedy,

It is a pleasure to accept your manuscript entitled "The unexpected narrowness of eccentric debris rings: a sign of eccentricity during the protoplanetary disc phase" in its current form for publication in Royal Society Open Science.

Kind regards,
Andrew Dunn

Royal Society Open Science Editorial Office
Royal Society Open Science
openscience@royalsociety.org

on behalf of Dr Leonardo Testi (Associate Editor) and Rob Ivison (Subject Editor)
openscience@royalsociety.org

Appendix A

15 May 2020

Dear Editor,

Thank you for sending on the referee's comments, they have been very useful. I am now submitting an updated version of the manuscript, which incorporates modifications in response to all of their suggestions. These are marked in bold face, as are a few minor changes.

Specific responses to the comments are appended below.

regards
Grant

--

Reviewer comments to Author:
Reviewer: 1

Comments to the Author(s)

This manuscript presents an interesting study on the radial width and eccentricity of a small sample of debris disks recently observed by ALMA. In particular, it focuses on the geometry of two well-studied systems, Fomalhaut and HD 202628, which show very narrow and eccentric rings. After deriving new constraints on the geometry of those two debris disks via model-fitting of the ALMA observational data, the author shows how these characteristics cannot be reconciled with the often-assumed scenario of secularly perturbed planetesimals that started with zero (or near-zero) initial eccentricities. Possible alternative scenarios that may explain the observed geometries are discussed, together with some ideas for future observations that may test the predictions from those scenarios. Although the available data does not allow the author to obtain conclusive results on the origin of these narrow and eccentric debris disks, the analysis presented here will be certainly useful for future similar investigations on the dynamics of these systems. For these reasons, I suggest to have this manuscript published after the authors address my comments listed below.

- My main comments refer to the presentation of the models used by the author.
More details should be provided both for the procedure adopted to run the N-body simulations discussed throughout the text, as well as on the method used to convert the results of the N-body simulations into images for comparison with the ALMA interferometric data.

> OK, there are a few steps, and I have spelled all of these out in a new paragraph. Not that it is a substitute, but the code that does this is available on github, and would help anyone actually wanting (or not wanting) to make their own version of this model. Note that there are no n-body simulations - since the behaviour due to secular perturbations is well approximated by the motion in eccentricity phase space, the models are entirely analytic (n-body models would be computationally impossible for MCMC fitting too I expect).

- Fig. 1: I would suggest to give more explicit definitions of all the parameters shown in Figure 1, for example inside the figure caption.

> Reviewer 2 suggested that this section could benefit from a bit of reorganisation, so I have attempted to include better definitions of all parameters as the model is described.

- When describing Fig. 1 in lines 10-12 of pag. 5, it is not immediately clear what the author means by "range of eccentricities" vs "dispersion of eccentricities".

> I was aiming to use "dispersion" as a technical term relating to a Gaussian distribution (i.e. in this case the dispersion of the Gaussian distribution of e_p), while range is more general (and could arise because of an increased dispersion, but in this case is because e_p is large and hence the full range of e is large, even though $\sigma_{e,p}$ is small). In rewriting parts of this section I have tried to make this distinction clear, and I have added a footnote for explicit clarification.

- Pag. 5, lines 45-47) It would be important to spell out the specific "reasonable" assumptions about the initial conditions.

> This was referring to the gas damping noted earlier in the same paragraph.

These statements are now essentially in the same place in the new version, so what "reasonable" means should be clearer.

- When discussing the ALMA data for both Fomalhaut and HD 202628, it would be important to at least state some of the key properties of those observations, e.g. wavelength and angular resolution.

> OK, these have been included.

- Fig. 8: it would be useful to add errorbars, when possible, to the datapoints shown on that plot.

> Yes, good point. These have been added.

- Pag. 13, line 43: "indicating that their width is greater": does the author mean "lower" instead?

> Yes!

- Fig. 9-14: I would suggest to increase the size of the characters, right now they are very difficult to read in the printed version.

> Indeed. Label sizes have been increased everywhere. In the appendix I also cut the panels for the disk geometry parameters Omega and i, and omega, adding the results for the former two to the text (and the latter is already discussed). This has allowed the font size to increase a bit.

- Typos:

Pag. 12, line 44: "acheiving"

Pag. 12, line 52: "eresoution"

Pag. 16, line 18: "epair shepherding planets"

> Fixed, thank you.

Reviewer: 2

Comments to the Author(s)

This manuscript describes a re-analysis of ALMA data on the Fomalhaut and HD

202628 debris rings with special attention to the ring's radial widths and their constraints on the orbits of the planetesimals/dust making up the rings; the results suggest that in each ring, the proper and forced eccentricities of the dust particles differ significantly. I recommend publication after some relatively minor points (listed below) are addressed.

1) The discussion of the Fomalhaut analysis notes that the "full" models improve the fit over the "uniform simple" model. While it's explained in the text that the inclusion of the proper eccentricity dispersion specifically helps to capture the disc radial profile, it could also be said that adding parameters to the fitted model should generally improve the fit. Is there a way to quantify the fit improvement and show that it's better than what would be expected simply from adding a parameter?

> In principle this can be done with multineest, though the fitting runs were not set up in a way to do this as model selection isn't important for the conclusions. Also, in part the goal of the full models was to use a more realistic model that would ensure the uncertainties on the other parameters weren't underestimated (e.g. by including a scale height, which wasn't included in the MacGregor paper). Regardless, for Fomalhaut a calculation using the Schwarz criterion (BIC), shows that compared to the simple model, the "penalty" term increases by about 45 for the three additional parameters in the full models, which is much less than the improvement in χ^2 of 240, so the additional parameters appear justified. I have added a few sentences stating this. The extra parameters are not justified for HD 202628; while this was previously implied, I have added a note to make it explicit.

2) I think the extra discussion of Fig 1 is a good idea given the use of similar figures later, but I think the presentation could be clearer, especially as it seems to be targeted toward readers who aren't very familiar with the dynamics. Eg, it isn't immediately clear from the text what the difference between "range" in eccentricity and eccentricity "dispersion" is (later in the text, the two terms are used somewhat interchangeably), or why two variables are needed to represent the "random component" of the

eccentricity. Perhaps the dynamics could be explained before describing the figure's different examples of this dynamics -- that way all the relevant variables are introduced completely and the reader has the tools to see what is going on in the figure when it's brought up for the first time.

> Fair point, the reason for the previous layout was that I find it very hard to describe the dynamics without the plots. However, describing the physics a bit first is probably better. To help with this I have added an illustrative figure at the start, and shuffled things so that the basic dynamics text comes before the plots with populations of particles. Most of the text is the same, but has changed enough that much of this section is bold face.

> Perhaps the confusion between range/dispersion is in part contributed by my use of it generally for semi-major axis distributions, since the models use both hard edges and a Gaussian distribution (so using dispersion not appropriate). I have tried to ensure the distinction between range and dispersion is clear, introducing and using dispersion as a technical term describing a Gaussian parameter, and range as a more general concept.

3) In section 4 and the later parts of section 3 it became clear that the usual expectation was that the forced and free eccentricities should be roughly equal, but I couldn't find an explanation/discussion of that expectation in the text.

> While this may be addressed by the updated text in S2, I have added an explicit statement noting that this is implied by small initial eccentricities.

4) The discussion of the prior Fomalhaut analysis motivates the current reanalysis pretty clearly, but it was less clear from the description of the HD 202628 prior work why a reanalysis was called for. While a comparison with the Faramaz 2019 paper shows that the modelling goals and procedures are rather different in the present work, it would be nice to add some discussion of the differences to the text in the beginning of section 3.

> The honest reason for reanalysis is that it's to ensure I trust and understand the results, e.g. it was only after all the modelling that I found the mistakes in both the MacGregor and Faramaz papers, which I might have missed otherwise (and obviously did miss as a co-author on the MacGregor

paper). For this paper however, it allows me to get the full posterior distributions for the parameters, which is useful to assess whether the proper and forced eccentricities are significantly different. While the conclusion on narrowness is the same as Faramaz+ for HD 202628, it's hard to assess how robust it would be to draw this conclusion purely from their paper. I have added a few sentences on this.

5) Finally, it would be great if the size of the text in the figure labels could be larger, to make the subscripts and appendix figures especially easier to read.

> Indeed. Label sizes have been increased everywhere. In the appendix I also cut the panels for the disk geometry parameters Ω and i , and ω , adding the results for the former two to the text (and the latter is already discussed). This has allowed the font size to increase a bit.